# Ultra-Rare Variants Identify Biological Pathways and Candidate Genes in the Pathobiology of Non-Syndromic Cleft Palate Only

**DOI:** 10.3390/biom13020236

**Published:** 2023-01-26

**Authors:** Emanuela Iovino, Luca Scapoli, Annalisa Palmieri, Rossella Sgarzani, Nayereh Nouri, Agnese Pellati, Francesco Carinci, Marco Seri, Tommaso Pippucci, Marcella Martinelli

**Affiliations:** 1Department of Medical and Surgical Sciences, University of Bologna, 40138 Bologna, Italy; 2Department of Experimental, Diagnostic and Specialty Medicine, University of Bologna, 40138 Bologna, Italy; 3Craniofacial and Cleft Research Center, Isfahan University of Medical Sciences, Isfahan 81745-319, Iran; 4Department of Translational Medicine, University of Ferrara, 44121 Ferrara, Italy; 5Medical Genetics Unit, IRCCS Azienda Ospedaliero-Universitaria di Bologna, 40138 Bologna, Italy

**Keywords:** non-syndromic cleft palate, nsCPO, risk factors, whole exome sequencing, gene set enrichment analysis, collapse analysis

## Abstract

In recent decades, many efforts have been made to elucidate the genetic causes of non-syndromic cleft palate (nsCPO), a complex congenital disease caused by the interaction of several genetic and environmental factors. Since genome-wide association studies have evidenced a minor contribution of common polymorphisms in nsCPO inheritance, we used whole exome sequencing data to explore the role of ultra-rare variants in this study. In a cohort of 35 nsCPO cases and 38 controls, we performed a gene set enrichment analysis (GSEA) and a hypergeometric test for assessing significant overlap between genes implicated in nsCPO pathobiology and genes enriched in ultra-rare variants in our cohort. GSEA highlighted an enrichment of ultra-rare variants in genes principally belonging to cytoskeletal protein binding pathway (Probability Density Function corrected *p*-value = 1.57 × 10^−4^); protein-containing complex binding pathway (*p*-value = 1.06 × 10^−2^); cell adhesion molecule binding pathway (*p*-value = 1.24 × 10^−2^); ECM-receptor interaction pathway (*p*-value = 1.69 × 10^−2^); and in the Integrin signaling pathway (*p*-value = 1.28 × 10^−2^). Two genes implicated in nsCPO pathobiology, namely COL2A1 and GLI3, ranked among the genes (n = 34) with nominal enrichment in the ultra-rare variant collapsing analysis (Fisher’s exact test *p*-value < 0.05). These genes were also part of an independent list of genes highly relevant to nsCPO biology (n = 25). Significant overlap between the two sets of genes (hypergeometric test *p*-value = 5.86 × 10^−3^) indicated that enriched genes are likely to be implicated in physiological palate development and/or the pathological processes of oral clefting. In conclusion, ultra-rare variants collectively impinge on biological pathways crucial to nsCPO pathobiology and point to candidate genes that may contribute to the individual risk of disease. Sequencing can be an effective approach to identify candidate genes and pathways for nsCPO.

## 1. Introduction

The human secondary palate develops between the fifth and twelfth embryonic weeks in order to separate the nasal and oral cavities. It originates from two distinct and symmetric palatal shelves that initially grow next to the tongue in a vertical position, then rotate to acquire a horizontal position on the top of the tongue, and finally, shelves approach and fuse with each other in an antero-posterior direction [1,2]. The cleft of the secondary palate is a common birth defect that may arise in any one of these developmental steps, possibly caused by failure of growth, elevation, adhesion, or fusion of the palatal shelves.

Secondary cleft palate can occur as an independent defect, a disorder known as non-syndromic cleft palate (nsCPO). However, in about 50% of cases, cleft palate occurs in conjunction with other abnormalities that contribute to recognizable or non-recognizable syndrome. In addition, a cleft palate is often associated with other orofacial clefts, possibly involving the lip, the alveolar ridge, or the primary palate; this is a different condition known as cleft lip, with or without cleft palate, that can be non-syndromic (nsCL/P) or syndromic.

nsCPO, as well as nsCL/P, are considered complex diseases caused by the interaction of several genetic and environmental factors. Indeed, epidemiological data support the multifactorial threshold model of inheritance [3]. The genetic architecture underlying nsCPO is still largely unknown. Several candidate loci have been proposed but have rarely been replicated in subsequent studies, leaving many hypotheses open. Convincing evidence supporting a role in nsCPO etiology was collected for FOXE1, GRHL3, and PAX7 (reviewed by Martinelli et al. [4]).

In recent years, genome-wide association studies (GWAS) have identified numerous single nucleotide polymorphisms and several genes involved in nsCL/P etiology. In contrast, this approach has not been as successful for nsCPO [5,6], although nsCPO heritability suggests a strong contribution of genetic factors [3]. Although several explanations are conceivable, one possible reason could be related to inherent limitations of GWAS; this approach has high efficiency in detecting the role of common variants having moderate effect, but is less effective in the case of rare and ultra-rare variants [7]. Rare variants, which are numerous in every human genome [8] and are known to play a notable role in genetic diseases and in complex disorders [9], might have a greater effect in the etiology of nsCPO than in nsCL/P. Of note, in the GWAS study conducted by Leslie et al., the risk variant rs41268753 detected in the GRHL3 locus was relatively uncommon in the general population (minor allele frequency 3.2%), at a value close to the efficiency limits of GWAS. This evidence supports the idea that low-frequency variants may have relevance in the etiology of nsCPO [6]. In this scenario, sequencing might be a better approach to identify rare causative variants. To this purpose, two different research groups performed whole exome sequencing (WES), focusing on nsCPO familial cases [10,11]. Hoebel and colleagues analyzed WES data from 16 nsCPO first-degree relatives from eight multiplex families. They did not identify any genes with recurrent deleterious mutations; however, they suggested potential candidate genes [10]. Moreover, Liu et al. investigated five affected individuals from a single family and found evidence for a role of ARHGAP29 in the pathogenesis of nsCPO [11].

In this investigation, WES and pathway analysis were combined to find genes harboring deleterious ultra-rare variants and biological pathways associated with nsCPO by gene set enrichment analysis (GSEA) in a nsCPO cohort and in unaffected controls. Moreover, we assessed that genes implicated in nsCPO pathobiology ranked among the most enriched in ultra-rare variants in the nsCPO cohort. 

The aim of this study was to examine the role of rare variants that are highly likely to be functionally damaging and/or pathogenic. We believe this may be a valuable resource for future investigations to define nsCPO etiology, a necessary step to develop prevention strategies and improve patient care.

## 2. Materials and Methods

### 2.1. Sample Collection

Following the approval by the Human Research Ethics Committee of Area Vasta Emilia Centro (Cod. CE: 14020/2017) and Ethics Committee of Isfahan University of Medical Sciences (IR.MUI.MED.REC.1399.178), we enrolled 30 Italian and 7 Iranian patients affected by cleft palate only as unique malformation (nsCPO). Exclusion criteria were the presence of other congenital anomalies or major diseases, exposure to known risk factors such as phenytoin and warfarin, and tobacco smoking and alcohol consumption during pregnancy. An additional inclusion criterion, only for the Iranian probands, was being offspring of a consanguineous mating, i.e., offspring of first- or second-degree cousins. Consanguinity may increase the risk factor of non-syndromic orofacial cleft, especially if rare autosomal recessive alleles are causally implicated. Nine out of 30 Italian and 2 out of 7 Iranian cases had one or more relatives (second, third, or fourth degree) reported as being affected by nsCPO. As control samples, we used 38 genomically unrelated parents from trios sequenced as part of the clinical routine at the Medical Genetics Unit, IRCCS AOU Bologna, Italy. WES data from these subjects were used as control, as their offspring were found to not be affected by orofacial cleft. Written informed consent was collected from all involved subjects at the time of recruitment at each clinical site, including patients, their parents, and/or additional informative on family members, when available.

### 2.2. Sequencing and Variant Detection

Genomic DNA of the 37 nsCPO cases were sequenced at Theragen Etex Bio (Kwanggyo Technovally, lui-dong Suwon, KOREA) (14 samples) or Novogene (Wan chai, Hong Kong) (23 samples), using the SureSelect XT Human All Exon V6 kit. Controls were sequenced at Macrogen (Seoul, Republic of Korea) using the SureSelect Human All Exon V6 (Agilent, Santa Clara, CA, USA). 

Raw reads were processed as described elsewhere [12,13,14]. In particular, cases’ and controls’ variants were called and recalibrated according to GATK “Best Practices” [15] and annotated using the Ensembl tool Variant Effect Predictor v.76 [16] against GENCODE [17] canonical transcripts on reference genome GRCh37, using dbNSFP (dbNSFP v.3.5a) [18], CADD (GRCh37 v1.4) [19], and dbscSNV (dbscSNV v1.1) plugins. Runs of Homozygosity (ROHs) were detected from sequencing data in Iranian and Italian cases, and controls using AUDACITY [20]. Systematic ancestral differences between cases and controls were evaluated using Principal Component Analysis (PCA) through EIGENSOFT. PCA was conducted on 57,215 common HapMap SNPs extracted from sequencing data. Sequencing quality was measured per-sample using the GATK Depth of Coverage (GATK v.3.8, DoC) utility to obtain the mean sequencing coverage and percentage of bases covered <20× across all GENCODE protein-coding exons and across all exons for each gene. To address the confounding effect due to an imbalance of coverage, we excluded the genes with uneven coverage in cases and controls using the following criteria (coverage harmonization): Any gene should have <20% of bases covered <20× in each sample;Any gene should not differ by >10% in mean coverage between cases and controls.

A schematic overview of the bioinformatics steps, the metrics used to decide whether to include samples and genes in the downstream analyses, and the filters used for selecting ultra-rare variants are summarized in Figure 1.

### 2.3. Gene-Set Enrichment Analysis (GSEA)

Because the analysis of ultra-rare variants in gene sets has more statistical power than a gene-per-gene analysis, we could use more stringent conditions for filtering variants in Gene-Set Enrichment Analysis (GSEA) than in Gene-based Collapsing Analysis (GCA). In particular, we limited GSEA analysis to ultra-rare variants absent in gnomAD database v2.1.1 with a CADD score ≥ 20 (Figure 1), while we included variants present in gnomAD at low minor allele frequency (MAF) (<0.01) and without CADD restrictions in GCA. GSEA was performed by up-loading lists of genes with ultra-rare variants separately, in cases and controls, on the ToppGene portal (https://toppgene.cchmc.org) accessed on 16 July 2022 [21]. *p*-values for each of the two groups resulted from Topp.Fun analysis corrected for multiple testing with Benjamini-Hochberg *p*-values of the terms for each GSEA category (Gene Ontology [GO], GO Molecular Function [GO:MF], and GO Cellular Function [GO:CF]). *p*-values of cases were compared to those of controls and of cases’ synonymous variants to confirm that the enrichment was observed only for protein-altering variants in nsCPO [22]. Gene pathway annotations were retrieved from the KEGG database.

### 2.4. Gene-Set Burden and Variants Analysis

A hypergeometric test was performed to assess whether genes relevant to nsCPO pathobiology (n = 25) preferentially achieved lower *p*-values (< 0.05 nominal *p*-values) in GCA compared to the rest of protein-coding genes. The list of genes relevant to nsCPO pathobiology was obtained by intersecting multiple lists derived from Online Mendelian Inheritance in Man (OMIM), Human Phenotype Ontology, Gene-Ontology, Genecard, and Malacard databases. OMIM and GenCards were searched by Boolean string: “cleft palate” OR “bifid uvula” OR “cleft uvula”; specific reference IDs were used in the other cases: HumanPhenotypeOntology HP:0000175, GeneOntology GO:0060021 and GO:1905748, MalaCards CLF027. The list of genes that are nominally enriched in ultra-rare variants in the nsCPO cohort was obtained by GCA to evaluate the per-gene rare variant burden in cases versus controls. In GCA, per-gene variant counts were performed in the whole cohort using an in-house Perl script. We assigned a binary variable to each subject based on absence/presence, respectively, of any number of variants per subject. The number of cases and controls with at least one variant was used to assess enrichment for ultra-rare variants in either group using Fisher’s exact test. The nominal significance level, to identify genes nominally enriched in ultra-rare variants in cases compared to controls, was 0.05.

Statistical tests were performed in R v 3.5.1. Clinical significance of ultra-rare variants was assessed by manual curation of their list according to the American College for Medical Genetics (ACMG) standards and guidelines [23], restricted to genes that had orofacial clefts in their OMIM clinical synopsis. Moreover, we investigated whether there was a significant difference in the number of ultra-rare homozygous variants due to reported consanguinity in families of Iranian nsCPO cases compared to Italian cases.

Segregation analysis in informative families was performed, when possible, by variant-site targeted PCR and Sanger sequencing, only for variants identified in genes with the nominally significant *p*-values from GCA. 

## 3. Results

### 3.1. Quality Checks 

After quality checks of sequencing data, two cases did not achieve per-sample coverage thresholds (Appendix A). As expected, PCA showed ample demarcation between Italian and Iranian samples, but suggested substantial clustering between cases and controls of the same geographic origin (Appendix A). As stated in the Methods section, GSEA has more power than GCA because it is applied to groups of genes instead of single genes. For this reason, although population stratification is not considered to be a crucial confounder for ultra-rare variants [24,25], we decided to apply PCA indications in GSEA, limiting this analysis to the 28 Italian cases and 38 controls. As GCA in this study is not used to carry out an association analysis, but rather to obtain a list of top-ranking genes for the Gene-Set burden test, we instead included all the samples passing quality checks and all variants with MAF ≤ 0.01 and no CADD restriction. 

### 3.2. GSEA Results

Using the most stringent conditions defined in the Methods section, ultra-rare variants identified 586 genes in cases and 708 in controls, in which most variants were missense across both groups. Molecular functions [GO:MF], KEGG pathways, and Cellular Function [GO:CF] terms were significantly enriched in gene sets identified by mutations in nsCPO cases. However, only a few terms showed exclusive enrichment in cases, i.e., there was no evidence of enrichment in controls or for gene sets identified by cases’ synonymous variants (Figure 2A). The most significantly enriched MF process was the cytoskeletal protein binding (GO:0008092) (BH corrected *p*-value = 1.57 × 10^−4^ for cases; *p*-value = 0.15 for controls; *p*-value = 0.25 for synonymous variants). Other significantly enriched MF included microtubule plus-end binding (GO:0051010), the protein-containing complex binding (GO:0044877), and the cell adhesion molecule binding (GO:0050839) (BH *p*-value = 1.06 × 10^−2^; 1.24 × 10^−2^; 1.24 × 10^−2^, respectively). Moreover, GSEA detected different pathways showing enrichment for genes with ultra-rare variants in cases (Figure 2B); among them were the ECM-receptor interaction pathway (BH *p*-value = 1.69 × 10^−2^) and the Integrin signaling pathway (*p*-value = 1.28 × 10^−2^) (Table 1).

### 3.3. Gene-Set Burden Test Results

While GSEA pointed to interesting pathways, which turned out to be significantly enriched ultra-rare variants in nsCPO cases, GCA ranked genes according to their enrichment of ultra-rare variants in cases compared to controls, and was instrumental in demonstrating that GCA top-ranking genes (that is, genes having the lowest *p*-values) were significantly over-represented in the list of genes relevant to nsCPO pathobiology [24,25]. All samples considered, there were 16,749 ultra-rare variants (16,230 missense, 348 stop-gained or -loss, 171 splice-acceptor or -donor) in 7496 genes, according to the quality criteria defined in the Methods section. Although no gene reached the genome-wide significance level of association (Table 2), two genes in the nsCPO pathobiology list, namely COL2A1 and GLI3, achieved a nominal level of significance in the association test (Fisher’s Exact Test, *p*-value < 0.05) for the presence of an excess of ultra-rare damaging variants in cases compared to controls (Table 2 and Table 3). Genes relevant to nsCPO pathobiology achieved lower *p*-values compared to the other protein-coding genes included in GCA analysis (hypergeometric test, *p*-value = 5.86 × 10^−3^).

### 3.4. Clinical Variants

An analysis guided by ACMG criteria revealed that four nsCPO cases (10.8% of the sample study) carried pathogenic (P) or likely pathogenic (LP) variants in genes causing autosomal dominant syndromes, with cleft palate in their clinical synopsis (Table 4). Four of the homozygous ultra-rare variants identified in Iranian cases were inside large ROHs, and one of those involves an autosomal dominant disease gene with cleft palate as a clinical feature (NM_198216.2:c.686-136A>T). The number of genes with homozygous ultra-rare variants in Iranian cases did not differ significantly from Italian cases (8 SNVs in 7 samples *versus* 19 SNVs in 27 samples; Wilcoxon signed rank test *p*-value > 0.05) (Appendix A), suggesting that no plausible recessive variant was implicated in nsCPO pathogenesis, despite the high consanguinity rate and the excess of genomic inbreeding of the Iranian cases compared to Italian samples (Appendix A).

Ultra-rare variants in clinically relevant genes were tested for segregation by Sanger sequencing in the cases and in their available relatives (Appendix A). Segregation analysis found that most of the ultra-rare variants were inherited from one of the healthy parents, suggesting incomplete penetrance. Interestingly, COL2A1 (NM_001844.5:c.2659C>T) missense P SNV was found to be de novo and the IRF6 (NM_006147.4:c.82T>G) missense LP SNV segregated with nsCPO in the family.

## 4. Discussion

During morphogenesis, cells need to divide, move, and interact with each other and the extracellular matrix (ECM). Different events must be coordinated in terms of timing and space in a succession of perfectly orchestrated steps. When the coordinated interplay of individual cell behavior, with regard to migration, adhesion, growth, differentiation, and apoptosis, is disrupted, it can lead to phenotypic alterations, such as orofacial clefts. During palatogenesis, palatine shelves modify their position from vertical to horizontal above the tongue, interface at the midline, adhere to each other, and fuse. Each step involves epithelial and mesenchymal cells that are subject to a dynamic remodeling that involves their cytoskeleton and cellular structures, named cell junctions.

Despite the huge efforts spent to clarify the causes leading to orofacial clefts, the genetic etiology of nsCPO has remained largely elusive. Recently, however, a large-scale sequencing study of 756 trio genomes implicated *de novo* mutations in the pathogenesis of orofacial clefts (OFC) [22]. Although only a minority of the cases in this study had nsCPO (58), they were found to have an overall significant excess of *de novo* mutations in protein-coding genes, and enrichment analysis pointed to gene sets involved in limb-bud formation and many craniofacial disease terms. Based on the hypothesis that ultra-rare variants might explain at least part of the heritability of nsCPO, we conducted a WES study to identify genes that individually and collectively contribute to nsCPO risk. Our results support this hypothesis in three ways:GSEA highlighted genes with ultra-rare variants cluster into gene ontology sets that are highly relevant to nsCPO pathobiology (cytoskeleton rearrangement, cell-adhesion, ECM-cell interaction);GCA identified two top-ranking genes (COL2A1 and GLI3) significantly overlapping in the list of genes relevant to nsCPO pathobiology;Pathogenic and likely pathogenic variants in genes, accounting for autosomal dominant orofacial cleft syndromes (CDON, COL2A1, IRF6, SNRPB), have been identified in four cases (10.8%).

Cytoskeleton, cell junctions, and ECM hold a network of proteins strictly involved in the dynamic morphogenetic events that characterize palate development. The transmembrane glycoproteins of the cell junctions are characterized by an extracellular binding site for transmembrane proteins of other cells, or ECM elements, as well as a cytosolic domain, that indirectly interacts with the cytoskeleton and is involved in the regulation of important signaling pathways regulation. Enrichment analysis strongly indicates that rare damaging variants preferably occur in genes coding for these groups of proteins. Some of them have been proposed as candidate genes for OFC malformations.

The active role of actin filaments in mesenchymal reorganization during palatogenesis has already been evidenced [26], as has actomyosin-dependent cell contractility [27]. Based on these observations, genetic association between variants in the MYH9 gene and orofacial cleft has been demonstrated in several studies [28,29,30]. Interestingly, GSEA of genes carrying mutations in our dataset identified higher concordance with the cytoskeletal protein binding gene ontology set (GO:0008092). Among these genes, additional myosin family members are included (MYH15, MYH2, MYH3, MYH4, MYO18B, MYO15A). In Tbx1-knockout palatal shelves, a gene profile analysis showed that myosin heavy chain 3 (Myh3) and nebulin (Neb) were downregulated [31]. Other genes included in the GO:0008092 list have been related in some way to the OFC onset. Alpha-parvin (PARVA), considered a candidate gene for nsCL/P, was demonstrated to be involved in cell proliferation and migration of human oral keratinocytes by RNAi silencing experiments [32]. Filamin B (FLNB) is an actin-binding protein that interacts with receptors and intracellular proteins that regulate cytoskeleton-dependent cell proliferation, differentiation, and migration [33]. Pathogenic variants in FLNB cause disorders presenting a spectrum of phenotypes, which sometimes includes cleft palate [34,35]. Maternal mutations at FLNB were suggested to influence the risk of cleft in the offspring based on an interaction between maternal gene and specific teratogens or fetal genes [36]. A study of environmental exposures and the effect of parent-of-origin on a CL/P cohort evidenced a possible association between SNPs in ANK3 and maternal smoking [37].

In our patients, we found a significant enrichment of ultra-rare mutations in genes encoding protein-containing complex binding (GO:0044877), which is a process crucial for several morphogenetic steps. Among these genes, the most studied was GAD1, which encodes glutamate decarboxylase-67 (GAD67), an enzyme with a main role in the γ-aminobutyric acid (GABA) metabolism. In mouse and rat models, Gad1 played a role in the normal development of the palate [38,39], while a study on the role of miRNAs in the regulation of CP genes found that variants in GAD1 significantly contributed to the human cleft palate phenotype [40]. An association study carried out on non-Hispanic white families revealed a significant association between nsCL/P and rs1046117 mapping on Fos proto-oncogene (FOS) [41]. In addition, the association of noggin (NOG) with nsOFC was confirmed by GWASs and confirmed in a recent meta-analysis [42,43,44].

Two genes were among the genes with nominal enrichment in the ultra-rare variant collapsing analysis and are also included in the list of genes relevant to nsCPO pathobiology: type II collagen alpha-1 chain (COL2A1) and glioma-associated oncogene family zinc finger 3 (GLI3). Interestingly, COL2A1 is listed in three enriched pathways identified by GSEA, whereas the GLI3 gene is found in the protein-containing complex-binding gene category of the ontology. COL2A1 is a fibrillar collagen typical of cartilaginous tissues and essential for normal embryonic skeletal development. A variety of mutations in the gene sequence can lead to a range of COL2A1-related disorders exhibiting a phenotypic spectrum that includes cleft palate [45]. The first evidence for COL2A1 involvement in the occurrence of nsCPO was provided by Nikopensius and colleagues, who found multiple haplotypes in COL2A1 (and COL11A2), associated with nsCPO, in an association study carried out on a cohort composed of 104 cases and 606 controls from the Baltic region [46]. GLI3 is one among the three members of the GLI family of zinc finger transcription effectors of the sonic hedgehog signaling pathway. GLI family members regulate gene expression and repression through the transduction operated by primary cilia in many tissues at various phases of embryonic morphogenesis. In particular, GLI3 mostly acts as a repressor. In 2008, Huang’s group correlated GLI3 deficiency with a high incidence of cleft palate, associated with abnormal tongue development, in a murine model [47]. In an association study conducted on 504 Chinese cases and 455 healthy controls, risk variants of GLI3 were identified as significantly associated with nsCL/P susceptibility [48]. The pivotal role played by primary cilium during the development of the orofacial region could be confirmed by the fact that more than 30% of ciliopathies are mainly defined by craniofacial defects, including lip/palate clefting.

Finally, we identified LP/P variants in genes associated with AD syndromes with cleft palate in their clinical synopsis. The nonsense variant in COL2A1 (NM_001844.5:c.2659C>T; p.Arg887Ter) was previously reported in Clinvar (Table 4), as implicated in Stickler syndrome [49,50]. In addition, the missense variant in IRF6 (NM_006147.4:c.82T>G; p.Trp28Gly) has been reported in ClinVar as variant of uncertain significance. In contrast, the stop-loss variant in SNRPB and the nonsense variant in CDON are novel findings and are not observed in the general population (gnomAD), nor have they been previously identified among patients.

Some interesting lines of discussion emerge from the points above. First, the present study confirms that nsCPO etiology is complex and also suggests that ultra-rare variants can act as susceptibility factors in nsCPO onset. Nonetheless, a portion of cases, albeit small, is attributable to monogenic causes of disease, being carriers of pathogenic or likely-pathogenic variants, strongly suggesting that ultra-rare variants contribute substantially to risk of nsCPO. Interestingly, the GCA and GSEA also support this view. The main limitation of this study was the small sample size, considering that nsCPO is a complex disease. In principle, the lack of Iranian controls may have affected the results of GCA analysis, but this is unlikely to have a tremendous impact because Iranian samples are a clear minority of the total cases. Nonetheless, indications of specific genes emerge quite clearly. COL2A1 is emblematic in this sense. Indeed, it is not only found in a case with a de novo pathogenic variant, but it is also among the genes with the lowest *p*-values in GCA and is represented in four significantly enriched gene sets. This is consistent with a previous report describing COL2A1 variants as a risk factor for nsCPO [46]. Similarly, MACF1 emerged as a potential candidate gene, in line with previous findings by Bishop et al. [22], as it is present in three of our significantly enriched gene sets. These gene sets are all implicated in biological functions or processes that are relevant to nsCPO etiology, such as cytoskeletal organization, cellular adhesion, and interaction with ECM, which are all crucial during palate morphogenesis.

Another limitation of this study is that Copy Number Variants (CNVs) were not included in the analysis. Although it is unlikely that, with the present small sample size, we could identify a CNV contribution, it has been recently suggested [51] that their role in orofacial clefts has been relatively overlooked.

In conclusion, WES analysis in patients with isolated cleft palate led to the discovery of ultra-rare variants in a number of genes already in the spotlight as candidate genes for orofacial cleft. We suggest that the approach is effective and allows us to identify relevant pathways in nsCPO pathobiology. Identification or confirmation of genes likely to contribute to the risk of nsCPO expands our knowledge of the genetic architecture and etiopathogenesis of this very common craniofacial birth defect. However, further investigations are needed in order to confirm the role of these genes in the development of the palate.

## Figures and Tables

**Figure 1 biomolecules-13-00236-f001:**
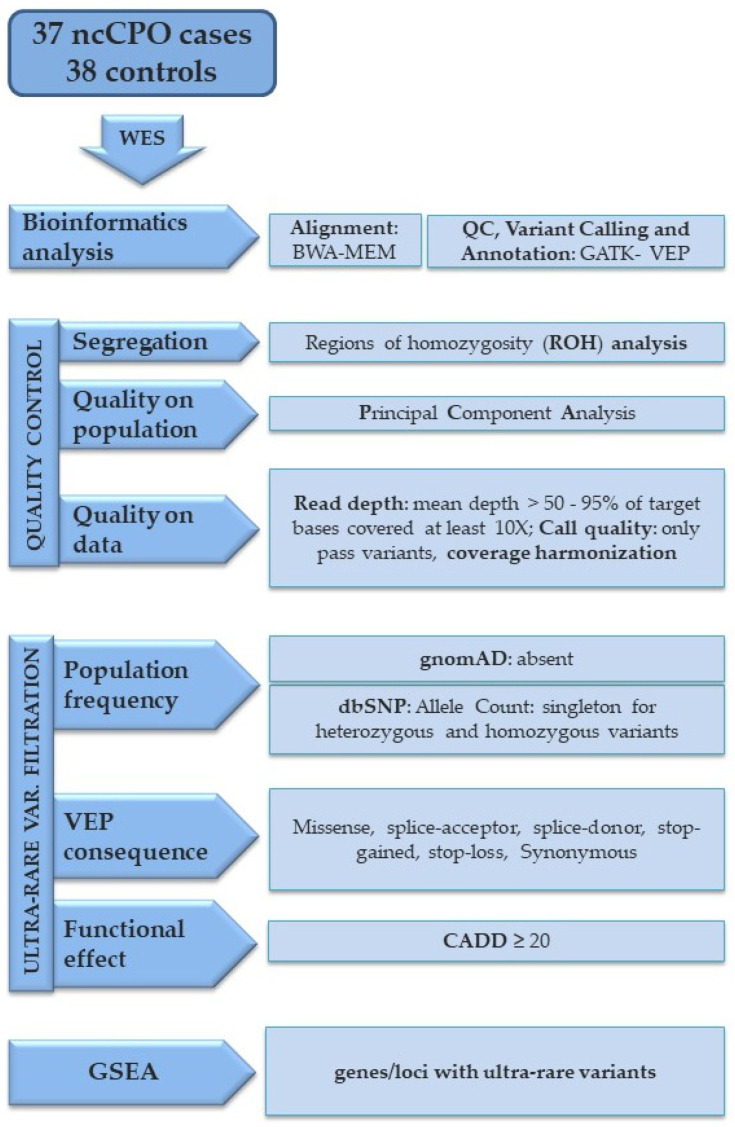
Flowchart of the analysis steps for generating gene lists with ultra-rare variants. The flowchart provides an overview of the approach used to select genes carrying ultra-rare variants from whole exome sequencing data for GSEA. Raw reads were aligned against hg19 reference genome using BWA MEM and, after duplicates were removed, SNVs were called using GATK. ROH analysis and PCA were performed as reported in the Methods section. Only the variants with a PASS filter were retained in the VCF file. Quality metrics were used to decide whether to include genes in downstream analysis; only genes with mean depth > 50× and 95% of target bases covered at least 10× were included. Moreover, genes with differences in coverage between cases and controls were excluded (see details for this in the Methods section). Variants predicted to be protein-altering (missense, splice-acceptor, splice-donor, stop-gained, stop-loss) were aggregated. Synonymous variants were only used in GSEA as a comparison group of variants, as they are not as likely to be implicated in nsCPO. Population frequency filter and functional effect predictor score were applied as described in the Methods section.

**Figure 2 biomolecules-13-00236-f002:**
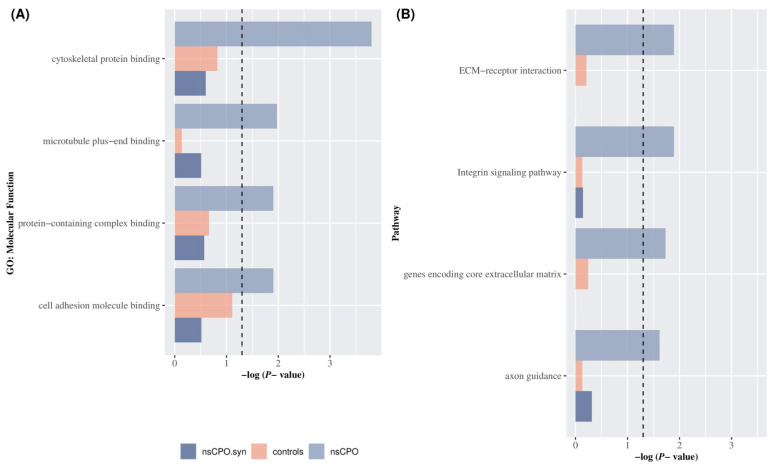
Gene Set Enrichment Analysis for ultra-rare variants in nsCPO and unaffected controls. GSEA for genes carrying protein-altering (lilac), synonymous (purple) ultra-rare variants in nsCPO cases, and protein-altering ultra-rare variants in unaffected controls (pink). The X-axis shows a negative log-transformed BH *p*-value, and the horizontal dashed line indicates the nominally significance threshold (*p*-value = 0.05). (**A**) BH *p*-value for genes with protein-altering and synonymous ultra-rare variants in nsCPO and protein-altering ultra-rare variants in unaffected controls for the top 5 significant Molecular Function (GO:MF) terms. (**B**) BH *p*-value for genes with protein-altering and synonymous ultra-rare variants in nsCPO cases and protein-altering ultra-rare variants in unaffected controls for the top 5 significant pathway terms.

**Table 1 biomolecules-13-00236-t001:** Genes carrying ultra-rare variants in nsCPO patients belonging to significant Gene Ontology categories identified by gene set enrichment analysis.

Category	Term	Nominal *p* *	Adjusted *p* *	Genes
GO: MF	cytoskeletal protein binding (GO:0008092)	1.43 × 10^−7^	1.57 × 10^−4^	RAB11B, XIRP1, ABL2, MAP1A, OBSCN, PARVB, ADGRV1, TUBGCP5, FMNL2, TTN, AP1G1, FSCN2, TAOK1, KIF17, UTRN, FKBP4, FLII, FLNB, VPS18, ANK3, CLIP2, APC, PPARG, CLSTN1, CLIP3, SETD2, PARVA, VPS41, MAST1, MYH15, MYBPC3, MYH2, MYH3, MYH4, TTLL9, EML2, PANX1, ACE, FRMD4B, IQGAP1, MAPK8IP3, TTBK2, CYFIP1, BAIAP2L1, NAV3, KCNQ2, MYO18B, GNB3, TBCD, KPNB1, NRCAM, SYNM, SYNE1, NUMA1, CLIP4, ARL4C, FARP1, MACF1, KIRREL1, DAAM2, RFLNA, MYO15A, TTLL13
GO: MF	microtubule plus-end binding (GO:0051010)	1.92 × 10^−5^	1.06 × 10^−2^	CLIP2, APC, CLIP3, TTBK2, NUMA1, CLIP4
GO: MF	cell adhesion molecule binding (GO:0050839)	3.97 × 10^−5^	1.24 × 10^−2^	RAB11B, ABCF3, CDHR3, TRPC4, OBSCN, FMNL2, TENM2, TAGLN2, UTRN, FLNB, ANK3, ITGA10, APC, NTNG1, AHSA1, TENM3, TNN, PARVA, TES, ADAM15, ITGB7, IQGAP1, PTPRB, PTPRH, BAIAP2L1, PTPRZ1, NRCAM, DOCK9, LAMA3, LAMB2, GRIN2A, GRIN2B, MACF1, KIRREL1, CDH7
GO: MF	protein-containing complex binding (GO:0044877)	4.52 × 10^−5^	1.24 × 10^−2^	XIRP1, NOG, ABL2, MAP1A, PDK2, TSHR, ARID1A, FMNL2, TTN, HIRA, ARID1B, AP1G1, FCGR2A, OTOF, AEBP1, MIF, FSCN2, PLCB2, CNGB1, UTRN, NCOA2, FLII, ANK3, ABI3BP, FOS, ANXA11, ITGA10, APC, CPD, CDC42BPB, PEX26, MTOR, TNN, PARVA, MYH15, CTSK, GAD1, MYH2, MYH3, MYH4, SPHK2, ADAM15, VPS33A, PANX1, ITGA2B, ITGB7, IQGAP1, ITPR1, MED24, CYFIP1, SACS, PTPRZ1, GLI3, GNB3, KPNB1, DOCK2, HNF1B, CARMIL2, SYNM, SYNE1, NUMA1, LAMA3, GRB2, LAMB2, LAMB3, GRIN2A, GRIN2B, LIN28A, RELL2, H1-0, H1-5, LRP2, MACF1, MYO15A, CPEB1
Pathway	ECM-receptor interaction (M7098)	7.03 × 10^−6^	1.69 × 10^−2^	COL2A1, COL4A2, COL5A2, ITGA10, TNN, LAMA1, ITGA2B, ITGB7, LAMA3, LAMA4, LAMB2, LAMB3
Pathway	Integrin signaling pathway (P00034)	1.07 × 10^−5^	1.28 × 10^−2^	PARVB, MAP3K4, COL2A1, COL4A2, COL5A2, FLNB, COL7A1, ITGA10, PARVA, MAP2K3, LAMA1, ITGA2B, ITGB7, LAMA3, LAMA4, LAMB2, LAMB3
Pathway	Ensemble of genes encoding core extracellular matrix including ECM glycoproteins, collagens and proteoglycans (M5884)	2.77 × 10^−5^	1.86 × 10^−2^	CRIM1, AEBP1, COL2A1, COL4A2, COL5A2, COL7A1, ABI3BP, COL28A1, EFEMP2, NTNG1, ZP1, SNED1, TNN, LAMA1, TINAGL1, IGSF10, PXDN, VWA5B1, LAMA3, LAMA4, LAMB2, LAMB3
Pathway	Axon guidance (1270303)	5.21 × 10^−5^	2.42 × 10^−2^	EPHB3, SCN11A, ERBB3, ABL2, RPS6KA2, TRPC4, SCN10A, FGF6, ALCAM, PLXNB1, RASAL2, COL4A2, ANK3, RASA3, CNTNAP1, ITGA10, RAPGEF2, PSMA5, LAMA1, FRS2, ITGA2B, IQGAP1, KSR1, KCNQ2, CACNB1, TRPC7, NRCAM, CAMK2G, KSR2, GRB2, SRGAP2, GRIN2A, GRIN2B, ROBO1

GO, gene ontology; * nominal and adjusted *p*-values of the GSEA analysis in cases.

**Table 2 biomolecules-13-00236-t002:** Genes obtaining nominal *p*-value < 0.05 in GCA.

	Cases		Controls		
GENES	Variants	No_Variants	Variants	No_Variants	*p*-Value
ZFYVE26	7	28	0	38	0.0041
THSD7B	6	29	0	38	0.0095
SPATC1	6	29	0	38	0.0095
IGHG1	6	29	0	38	0.0095
EXO1	0	35	7	31	0.0119
JAKMIP3	9	26	2	36	0.0210
AHNAK	9	26	2	36	0.0210
PLEKHN1	5	30	0	38	0.0216
NBEAL2	5	30	0	38	0.0216
USP42	5	30	0	38	0.0216
OR4K1	5	30	0	38	0.0216
STARD9	5	30	0	38	0.0216
UNC13A	5	30	0	38	0.0216
OLFML2B	4	31	0	38	0.0481
MROH9	4	31	0	38	0.0481
SLC4A5	4	31	0	38	0.0481
VWA3B	4	31	0	38	0.0481
CAND2	4	31	0	38	0.0481
USP4	4	31	0	38	0.0481
LRP2BP	4	31	0	38	0.0481
ERAP2	4	31	0	38	0.0481
DOPEY1	4	31	0	38	0.0481
GLI3	4	31	0	38	0.0481
TRBV6-7	4	31	0	38	0.0481
LOXL2	4	31	0	38	0.0481
APOBEC1	4	31	0	38	0.0481
COL2A1	4	31	0	38	0.0481
LIMA1	4	31	0	38	0.0481
HAL	4	31	0	38	0.0481
RAI1	4	31	0	38	0.0481
MPP3	4	31	0	38	0.0481
RSAD1	4	31	0	38	0.0481
CILP2	4	31	0	38	0.0481
ZNF600	4	31	0	38	0.0481
LILRA4	4	31	0	38	0.0481
TUBB1	4	31	0	38	0.0481
UTRN	6	29	1	37	0.0497
NLRC3	6	29	1	37	0.0497
URB1	6	29	1	37	0.0497

**Table 3 biomolecules-13-00236-t003:** Variants identified by GCA in COL2A1 and GLI3 are detailed.

Gene	Genomic Position *	Variant Type	Variant Class	CADD Score	Protein Consequence	Population Frequency	Population
COL2A1	12-48377883-G-T	SNV	missense	28	p.Pro643His	7.16 × 10^−6^	Iranian
COL2A1	12-48369250-C-T	SNV	missense	9	p.Gly1246Ser	0.000598	Italian
COL2A1	12-48373812-G-A	SNV	Stop gain	46	p.Arg887Ter	0	Italian
COL2A1	12-48376305-C-T	SNV	missense	19	p.Ala761Thr	8.77 × 10^−6^	Italian
GLI3	7-42007446-C-T	SNV	missense	27.5	p.Gly727Arg	0.007131	Italian
GLI3	7-42063171-C-G	SNV	missense	23.5	p.Gly465Arg	0.003811	Iranian
GLI3	7-42187851-C-T	SNV	missense	28.4	p.Arg114Lys	0.002434	Italian
GLI3	7-42187885-G-A	SNV	missense	24.7	p.Pro103Ser	4.03 × 10^−6^	Italian

* Human assembly GRCh37/hg19; CADD score, Combined Annotation Dependent Depletion score.

**Table 4 biomolecules-13-00236-t004:** Pathogenic and likely pathogenic variants of cases identified in genes associated with syndromes involving CPO.

Genes	Genomic Position *	Protein Consequence	HGVS Nomenclature	ACMG Classif.	OMIM Disease and Inheritance	ClinVar Accession	Genotype
CDON	11-125875899-G-A	p.Gln536Ter	NM_001243597.2: c.1606C>T	LP	Holoprosencephaly 11; AD		Het
COL2A1	12-48373812-G-A	p.Arg887Ter	NM_001844.5: c.2659C>T	P	Stickler syndrome, type I; AD	VCV000817513.4	Het
IRF6	1-209974677-A-C	p.Trp28Gly	NM_006147.4: c.82T>G	LP	{Orofacial cleft 6}; AD	VCV000464464.1	Het
SNRPB	20-2442575-T-A	p.Ter232Cysext *?^§^	NM_198216.2: c.686-136A>T	LP	Cerebrocostomandibular syndrome; AD		Hom

* Human assembly GRCh37/hg19; ^§^ StopLoss variant: the stop codon (Ter/*) variant at position 232 converts it into Cys codon and adds a tail of new amino acids of unknown length (position * ?). HGVS, Human Genome Variation Society; ACMG classif., American College for Medical Genetics classification; OMIM, Online Mendelian Inheritance in Man database; P, pathogenic; LP, likely pathogenic; AD, autosomal dominant; Het, heterozygous; Hom, homozygous.

## Data Availability

The datasets presented in this study can be found in an online repository. The name of the repository and accession number can be found below: https://www.ncbi.nlm.nih.gov/sra/PRJNA870708.

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
