# Peer review of "Ultra-Rare Variants Identify Biological Pathways and Candidate Genes in the Pathobiology of Non-Syndromic Cleft Palate Only"

_biomolecules, 2023, doi:10.3390/biom13020236_

Round 1

Reviewer 1 Report

The manuscript “Ultra-rare variants identify biological pathways and candidate genes in the pathobiology of non-syndromic Cleft Palate Only” is a clear and well-organized study of genetic contributions to nsCPO in a sample of nsCPO patients and controls. The authors use generally accepted methods to isolate rare and ultra-rare variants through whole exome sequencing and discuss the known functions of the genes in which variants were identified and their impacts on palatal morphogenesis.

I do not have the necessary expertise to critically evaluate the details of the methodology, so my review is focused on other aspects of the manuscript. This is an important topic of study given the heterogeneity of nsCL/P, and the identification of ultra-rare variants not previously studied in this context could help researchers and clinicians develop new models and new tools for assessing risk in patients going forward.

General concept comments

I really appreciated the discussion of the variants identified, but I was left wondering whether any of the ultra-rare variants seen in this sample have not been implicated in nsCLO previously. An additional paragraph in the discussion focused on novel findings could strengthen the manuscript.

I am also left wondering whether the specific phenotypes of the patients could be assessed and incorporated into these analyses, given the heterogeneous presentation of nsCLO. If possible, this could be the subject of future research.

Overall, I found this to be a well-structured study of an important topic in nsCLO research.

Specific comments

There are many minor grammatical errors throughout the text, for example on lines 51, 52, 58, 63, 75, 77, 81, 85. The manuscript will require additional proofreading to fix these errors. One common error is overuse of “the”, especially at the beginning of sentences.

The inclusion criterion “being offspring of a consanguineous mating” is not justified in the Methods section, though it is analyzed and discussed throughout the manuscript. It would be helpful to explain this criterion in more detail, particularly since it was only used in one population and not the other and there may be other confounding differences between these populations. Did the Italian samples get tested for Runs of Homozygosity as well? Are the Italian samples definitely not consanguineous? What degree of consanguinity would meet this criterion? The explanation provided in lines 225-226 could be provided in the Methods section as well, but this will need to be explained further for readers less familiar with these methods.

Unclear what “when informative and available” means on lines 98-99…. Was informed consent always obtained from patients and parents? Was it only collected from additional family when informative? This should be clarified.

Is there any concern with sequencing samples at multiple sites and with slightly different kits for the affected and control samples? Was there any consistent difference in coverage observed between the kits?

On lines 185-186, “initially carried out GSEA and GCA on 28 Italian cases and 38 controls only.” Then, on line 225, “This analysis included both Italian and Iranian cases…”. It is unclear which analysis is being described on line 225. Is it referring to the description in the following paragraph? If so, this should be clarified on line 225 because the current phrasing seems to refer to the analyses described in earlier paragraphs.

For Figure S1, I recommend scaling the PCA axes so that the distance from -0.2 to 0.5 is approximately the same on PC1 and PC2. Right now the PC2 axis is much shorter than PC1 and implies that there is less variation on PC2 than is accurate.

Reviewer 2 Report

1、The necessity of ultra-rare variants detection should be described in more detail.

2、The screening methods and standards of ultra-rare variants  should be clarified in the research.

3、Reorganize the method and result section and add a heading to each part.
